# Effects of GABA on Oxidative Stress and Metabolism in High-Glucose Cultured Mongolian Sheep Kidney Cells

**DOI:** 10.3390/ijms251810033

**Published:** 2024-09-18

**Authors:** Rina Su, Longwei Chang, Tong Zhou, Fanhua Meng, Dong Zhang

**Affiliations:** 1College of Life Sciences, Inner Mongolia Agricultural University, Hohhot 010018, China; 17614832879@163.com (R.S.); clw201114@163.com (L.C.); zhtong0217@163.com (T.Z.); 2Inner Mongolia Autonomous Region Key Laboratory of Biomanufacturing, Hohhot 010018, China

**Keywords:** Mongolian sheep, renal cortex, renal medulla, high glucose, oxidative stress, untargeted metabolome

## Abstract

The Mongolian sheep, emblematic of the Inner Mongolian grasslands, is renowned for its exceptional stress resistance and adaptability to harsh environments, drawing considerable attention. Recent research has unveiled the novel role of γ-aminobutyric acid (GABA) in combating oxidative stress. This investigation examined how GABA impacts renal-cortex and medulla cells from Mongolian sheep exposed to high-glucose stress conditions, utilizing gene expression analysis and non-targeted metabolomics. Elevated glucose levels significantly reduced the viability of Mongolian sheep renal cells and increased reactive oxygen species (ROS) levels. Conversely, the introduction of GABA notably enhanced cell viability, reduced ROS production, and stimulated the expression of antioxidant genes (e.g., *Gpx*, *SOD*, *CAT*) in the renal cortex. In the renal medulla, *CAT* expression increased, while *Gpx* gene expression showed mixed responses. Metabolomics analysis indicated that high-glucose exposure altered various metabolites, whereas GABA alleviated the metabolic stress induced by high glucose through modulating glycolysis and the tricarboxylic acid cycle. In Mongolian sheep renal cells, GABA effectively mitigated oxidative damage triggered by high-glucose stress by upregulating antioxidant genes and regulating metabolic pathways, revealing insights into its potential mechanism for adapting to extreme environments. This finding offers a fresh perspective on understanding the stress resilience of Mongolian sheep and may provide valuable insights for research across diverse disciplines.

## 1. Introduction

The ability of animals to endure and thrive in challenging conditions stands as a testament to the remarkable resilience and perseverance inherent in living organisms. In hostile ecological settings, animals demonstrate remarkable fortitude by adapting swiftly to their environments. Mongolian sheep, a prominent livestock breed in Inner Mongolia, possess notable attributes such as robust vitality, adaptability to nomadic lifestyles, resilience to cold and arid climates, and high-quality meat and fat production capabilities. Regarded as a precious genetic reservoir for livestock, Mongolian sheep are esteemed as a national asset. The distinctive physiological and metabolic traits of Mongolian sheep make them a significant focus for investigations into animal resilience.

Oxidative stress (OS) is a fundamental concept in the life sciences and medical fields, initially outlined by Helmut Sies [1] and further explored through subsequent research. It represents an imbalance between the production of ROS and the body’s natural antioxidant defense mechanisms, playing a crucial role in the development of various diseases. ROS, acting as signaling molecules, are essential for maintaining cellular redox balance and regulating signaling pathways [2]. However, excessive ROS production overwhelms the antioxidant system, leading to oxidative stress that can jeopardize cell stability and overall organism health. Direct measurement of ROS levels is challenging due to their high reactivity and short lifespan [3], prompting researchers to focus on stable oxidative metabolites induced by ROS as indirect markers of oxidative stress. These metabolites’ content and distribution patterns offer a more accurate assessment of the oxidative stress status within organisms [4]. Based on their distinct chemical compositions and origins within biological systems, oxidative metabolites can be classified into three main groups: lipid peroxidation products [5,6], which primarily stem from the oxidative degradation of cell membrane lipids; protein oxidation products [7], which arise from the oxidation of amino acid residues within protein structures; and DNA oxidation products [8,9], which represent specific alterations induced by oxidative stress on DNA bases or deoxyribose segments. Furthermore, the quantification of ROS levels can be inferred through indirect methods such as evaluating intracellular antioxidant enzyme performance, oxygen-free-radical elimination rates, and related markers.

GABA was initially identified in saprophytic fungi in 1910, and subsequent investigations have confirmed its presence in various organisms [10,11]. GABA is predominantly synthesized in the cytoplasm and metabolized in the mitochondria [12,13,14]. Its distribution in ruminants is widespread [15], particularly in the brain and neurons, where it is most concentrated. GABA functions as a neurotransmitter through GABA type-A and GABA type-B receptors [16], not only within the central nervous system but also in the peripheral nervous system and other bodily organs. GABA plays a multifaceted role in modulating kidney function [17]. It has been observed to have various effects, including widening renal blood vessels, enhancing renal blood flow, optimizing the regulation of glomerular blood flow, inhibiting constriction, and dilating blood vessels. These actions contribute to improved efficiency in the removal of metabolic waste. Additionally, GABA influences the reabsorption function of renal tubular cells, aiding in the maintenance of water and salt balance in the body and indirectly influencing renal tubular function. Its antioxidant and anti-apoptotic properties offer protective benefits to renal tissues. Furthermore, GABA facilitates uric acid excretion and helps maintain uric acid metabolism equilibrium. During kidney development, GABA plays a vital role in promoting cell proliferation and differentiation [18,19], which are essential for kidney maturation.

The utilization of untargeted metabolomics in antioxidant research has garnered significant attention due to advancements in metabolomics technology. In recent years, untargeted metabolomics techniques, combined with LC-MS or GC-MS methodologies, have been extensively employed for the analysis of samples from various sources, including animals, plants, and microorganisms. Through comprehensive bioinformatic analyses, metabolites associated with antioxidants have been successfully identified, addressing a range of research inquiries [20]. For instance, a study investigating the metabolic profiles and antioxidant properties of different genotypes of Vitis vinifera at varying developmental stages revealed that distinctions between fruit developmental stages were more pronounced than those between genotypes. The antioxidant capacity was found to be positively correlated with specific nutritional biomarkers, offering valuable insights into the development of novel varieties with enhanced nutritional benefits [21]. Furthermore, the application of untargeted metabolomics to examining the evolving metabolic composition of the supraspinatus tendon in diabetic rats at different stages has provided novel perspectives on the mechanisms underlying diabetes-induced tendon pathology. Notably, the discovery of seven new metabolites, including uric acid and xanthine, has opened up potential avenues for further research in this area [22]. Metabolomics has also played a pivotal role in enhancing our understanding of the pathogenesis of chronic kidney disease (CKD) with the identification of novel biomarkers for early CKD diagnosis through metabolomic analyses of serum and urine samples from CKD patients.

In challenging environmental conditions, the kidney, a vital component of the excretory system in animals, is crucial for the adaptive processes observed in Mongolian sheep. However, investigating the kidney’s role in animal adaptation directly under in vivo or in vitro conditions poses difficulties. While individual-level multi-omics analysis techniques have been employed to study the adaptive mechanisms of the kidney [23], the intricate coordination of various organ systems at the whole-body level creates challenges in understanding the molecular mechanisms underlying the kidney’s unique functions. Culturing kidney cells in vitro offers a means to observe changes in cellular metabolic processes under different culture conditions, thereby providing fundamental insights into the kidney’s mechanisms. Limited research has explored the use of exogenous drugs to treat kidney damage in cultured kidney cells. Previous laboratory investigations into the cold resistance of Mongolian sheep revealed an observed inward transport of GABA in liver cells during exposure to cold stimuli, suggesting a potential physiological role of GABA in cold environments [24]. Furthermore, studies on the salt and drought resistance of camels have identified the gene encoding the GABA transporter protein *SLC6A1* as being associated with antioxidant effects in the Bactrian camel, indicating the potential involvement of GABA in responding to oxidative stress [25]. Drawing on previous research on GABA, this investigation involved the cultivation of kidney cells from two species under high-glucose conditions, followed by the addition of GABA and subsequent assessment of ROS levels and cell viability across various treatment groups. This study employed fluorescence-based quantitative PCR to identify variations in the expression of common antioxidant genes and performed non-targeted metabolomics analysis on the cortex and medulla cells of Mongolian sheep kidneys. By analyzing the distinct metabolites present in each group, the study postulated that GABA may offer protective benefits to the kidney cortex and medulla cells under conditions of heightened glucose stress. The examination of stress resistance in Mongolian sheep not only enhances our understanding of the adaptive mechanisms of these animals but also provides valuable insights and guidance for addressing challenges such as natural calamities and environmental fluctuations.

## 2. Results

### 2.1. Establishment and Maintenance of Primary Renal-Cortex and Medulla-Cell Cultures

Following a 4 h incubation period in a consistent temperature environment, tissue blocks of the renal cortex and renal medulla became attached to the surface. Subsequently, over a span of 3 days, a multitude of round and spindle-shaped cells emerged surrounding the renal-cortex and renal-medulla tissue blocks (Figure 1A), with predominantly spindle-shaped cells observed crawling out around the renal-medulla tissue blocks (Figure 1D). By the 7th day, there was a notable proliferation of cells in both the renal-cortex (Figure 1C) and renal-medulla regions (Figure 1F), achieving a confluence rate of 70–80% at the base of the culture dish. Notably, renal-cortex cells exhibited a faster growth rate within the initial five days compared to renal-medulla cells. Following cell passage, both types of cells adhered to the surface within approximately 2 h. The initial seeding quantity of renal-cortex and renal-medulla cells in the flask ranged between 200,000 and 300,000 cells, and after 96 h, both cell types had reached a confluence rate of 90% at the base of the flask (Figure 1G,H).

### 2.2. Effect of GABA at Different Concentrations on Renal Cells Cultured in Normal and High-Glucose Conditions

Under typical sugar levels, varying concentrations of GABA do not significantly affect the morphology and growth rate of kidney cells in the P3 generation. However, a concentration of 15 mmol/L GABA demonstrates a significant increase in the quantity of dead cells (Figure 2D). Conversely, in conditions of elevated sugar levels, the control cohort exhibits alterations in cell morphology and reduced density. In contrast, the group exposed to 10 mmol/L GABA displays the highest cell density and the lowest incidence of cell death (Figure 2F–H).

### 2.3. Results of ROS-Level Measurement

In the presence of normal sugar levels, an increase in GABA concentration leads to a rise in ROS levels in kidney cells, as depicted in Figure 3A–D,I. Specifically, when GABA concentrations reach 10 mmol/L and 15 mmol/L, there is a significant elevation in ROS levels compared to the control group (0 mmol/L), suggesting that high GABA concentrations may stimulate ROS production in a normoglycemic environment. Conversely, under high-sugar culture conditions, the introduction of GABA exerts an inhibitory influence on ROS levels in kidney cells. As GABA concentration rises, there is a decline in ROS expression, as illustrated in Figure 3E–I. Notably, when GABA concentrations are at 5 mmol/L and 10 mmol/L, there is a significant reduction in ROS expression compared to the control group.

### 2.4. Cell Viability Assessment Using CCK-8 Assay

Under normal sugar concentrations, an increase in GABA concentration promotes the enhancement of cell vitality, especially at 15 mmol/L (Figure 4A). However, under high-sugar conditions, cell vitality exhibits a trend of first increasing and then decreasing. Specifically, 5 mmol/L and 10 mmol/L of GABA significantly enhance cell vitality, whereas 15 mmol/L may have an inhibitory effect. Additionally, high sugar cultivation significantly increases ROS levels and decreases cell vitality, indicating that high concentrations of glucose have a damaging effect on kidney cells (Figure 4B,C). It is worth noting that the addition of GABA can reduce ROS production and restore cell vitality, with 10 mmol/L exhibiting the most favorable effect. Based on the above results, in order to minimize the potential interference of GABA on cell activity and ROS production, subsequent studies selected 10 mmol/L GABA as the treatment concentration.

### 2.5. Effect of 10 mmol/L GABA on ROS Levels in Renal-Cortex Cells Cultured in High Glucose

Figure 5A–D illustrates the impact of varying glucose concentrations on the growth of renal-cortex cells when treated with 10 mmol/L GABA. Exposure to high-glucose levels resulted in alterations in cell morphology and an increase in cell death rates, as depicted in Figure 5C. Conversely, the addition of GABA in a high-glucose environment maintained stable cell morphology, reduced cell death rates, and enhanced cell density, as evidenced in Figure 5D. The fluorescence images in Figure 5E–H display the detection of ROS in renal-cortex cells subjected to different sugar concentrations and GABA treatment. ROS fluorescence was observed in all experimental groups, with no significant disparity in average fluorescence intensity between the normal-sugar-plus-GABA group and the normal-sugar control group, as shown in Figure 5I. In contrast, the average fluorescence intensity in the high-sugar-plus-GABA group was notably lower than that in the high-sugar control group, indicating a substantial decrease in ROS production.

### 2.6. Effect of 10 mmol/L GABA on ROS Levels in Renal-Medullary Cells Cultured in High Glucose

Different glucose concentrations were used in combination with 10 mmol/L GABA to treat renal-medullary cells, and their growth conditions are shown in Figure 6. From the figure, it can be observed that there was no significant difference in cell status between the normal-sugar control group and the group with added GABA (Figure 6A,B); both the high-sugar control group and the group with added GABA had cells growing in the dishes (Figure 6C,D). Fluorescence images of ROS detection in renal-medullary cells treated with GABA at different sugar concentrations are presented in Figure 6E–H; fluorescence was observed in all groups, indicating varying degrees of ROS production in each group. The average fluorescence values for each group are calculated and presented in Figure 6I. Compared to the normal-sugar control group, the average fluorescence value significantly increased when GABA was added to renal-medullary cells cultured in normal-sugar conditions. For renal-medullary cells cultured in high-sugar conditions, the average fluorescence value significantly decreased when GABA was added, compared to the high-sugar control group, indicating a significant reduction in ROS production.

### 2.7. Functional Assessment of Renal Cells Cultured in High Glucose with 10 mmol/L GABA

Various sugars were introduced to the Mongolian sheep kidney cortex and medulla cells treated with GABA, and the results of the cell viability assessment are illustrated in Figure 7. The comparison revealed no notable difference in cell viability between kidney cortex and medulla cells treated with regular sugar and GABA, as compared to the control group treated solely with regular sugar. Nevertheless, a substantial enhancement in cell viability was observed when high sugar was combined with GABA-treated cells, in comparison to the control group treated only with high sugar, suggesting that the addition of GABA augmented the functionality of kidney cortex and medulla cells.

### 2.8. Gene Expression Changes in Renal Cells Treated with 10 mmol/L GABA in High-Glucose Conditions

The study investigated the expression of *SLC6A1*, *Glut1*, *SOD1*, *SOD2*, *Gpx1*, *Gpx4*, and *CAT* genes in renal-cortex and renal-medulla cells exposed to normal and high-glucose concentrations in the presence of GABA. The findings presented in Figure 8 demonstrate that in the renal cortex, the combination of normal glucose and GABA led to a decrease in *SLC6A1* expression, whereas high glucose with GABA resulted in an increase. Conversely, no significant alteration in *SLC6A1* expression was observed in the renal medulla. *Glut1* expression in the renal cortex was elevated under both normal and high-glucose conditions when combined with GABA. *SOD1* expression was increased with normal glucose and GABA, while high glucose with GABA notably affected both *SOD1* and *SOD2* expressions. *Gpx1* and *Gpx4* levels were elevated under both normal and high-glucose conditions with GABA, although *Gpx1* expression decreased in the renal medulla under high-glucose and GABA conditions. *CAT* expression increased in both the renal cortex and medulla upon the addition of GABA. These results indicate that GABA exerts intricate regulatory effects on cellular transport and antioxidant systems.

### 2.9. PCA Analysis of Metabolomics Data from Renal Samples

The study is structured into four distinct groups: LG1 (standard sugar without GABA), LG2 (standard sugar with 10 mmol/L GABA), HG1 (high sugar without GABA), and HG2 (high sugar with 10 mmol/L GABA), focusing specifically on the renal cortex. Additionally, there are MLG1, MLG2, MHG1, and MHG2 groups targeting the renal medulla, with each group having three replicates. The findings presented in Figure 9 demonstrate minimal systematic discrepancies among the groups, indicating stable experimental data and good repeatability. These results provide a dependable foundation for subsequent analysis.

### 2.10. Identification of Differential Metabolites in Renal-Cortex Cells

Metabolites that have a VIP value > 1 and a *p*-value < 0.05, as determined by Student’s *t*-test, are deemed to be significantly different. The volcano plot illustrating differential metabolites for each group is presented in Figure 10, revealing a lower number of differential metabolites in the comparison between LG2 and LG1. Detailed results of metabolite detection, identification, and the relevant differential metabolites can be found in Appendix A within this study.

### 2.11. Identification of Differential Metabolites in Renal-Medullary Cells

Figure 11 displays the volcano plot illustrating the variance in metabolites among the different groups. The presence of distinct metabolites in each group is evident. For detailed information on metabolite detection, identification, and the specific results of the differential metabolite analysis pertinent to this research, please consult Appendix A.

### 2.12. Construction of Metabolic Networks Based on Differential Metabolites

This research integrated the gene expression patterns of antioxidant-related genes with metabolomics analysis to establish a GABA-mediated antioxidant regulatory network model in renal cells (Figure 12). In conditions of elevated glucose levels, there was an augmentation in the co-transport of GABA and glucose, facilitated by the upregulation of *Glut1* and *SLC6A1* genes, enhancing their cellular absorption. Consequently, crucial intermediate metabolites of the glycolytic pathway (such as glucose-6-phosphate, fructose-6-phosphate, and glyceraldehyde-3-phosphate) were upregulated, intensifying glucose metabolism flow, resulting in the accumulation of NADH and FADH_2_ and potentially enhancing ROS production in the tricarboxylic acid cycle. In reaction to this, cells activated various antioxidant mechanisms: *SOD2* within mitochondria catalyzed the conversion of ROS to H_2_O_2_, which was subsequently broken down by *Gpx1*, while *SOD1* in the cytoplasm transformed ROS into H_2_O_2_, and *CAT* and *Gpx4,* respectively, decomposed H_2_O_2_ and directly eliminated ROS. Concurrently, the pentose-phosphate pathway was stimulated, leading to elevated levels of glucose-6-phosphate and ribose-5-phosphate and enhancing NADPH production, which acted as a crucial reducing agent in the glutathione cycle, reinforcing non-enzymatic antioxidant protection and ultimately effectively mitigating oxidative stress induced by high glucose levels.

## 3. Discussion

In recent years, GABA, a potent stress regulator in animals, has garnered increasing attention for its pivotal role in mitigating oxidative stress. This research investigates the impact of GABA on the renal-cortex and medulla cells of Mongolian sheep under normal-glucose-concentration (5.5 mmol/L) and high-glucose-concentration (25 mmol/L) conditions. This study establishes control groups for both glucose levels and examines the effects of GABA on cell viability and ROS levels. Results indicate that GABA supplementation significantly reduces ROS production and enhances cell viability under high-glucose conditions, while no significant physiological changes are observed under normal glucose levels. These findings suggest a potential antioxidant role for GABA in mitigating oxidative stress induced by high glucose levels. Further analysis reveals that, under high-glucose conditions, the expression of *SLC6A1* (encoding GABA transporter protein 1) increases, along with the upregulation of antioxidant genes (*SOD1*, *SOD2*, *CAT*, *Gpx1*). This suggests that GABA may enter cells via *SLC6A1*-mediated transport, promoting antioxidant gene expression and reducing ROS levels. Conversely, under normal glucose levels, the expression of *SLC6A1* decreases in renal-cortex cells after GABA supplementation, while no significant change is observed in medulla cells. This indicates that, under normal physiological conditions, Mongolian sheep renal cells may not require additional GABA supplementation for antioxidant effects, or there may be alternative, unidentified antioxidant mechanisms at play. Furthermore, this investigation identified notable variations in the responses of renal-cortex and renal-medulla cells during the antioxidation process. In particular, following the introduction of GABA under conditions of elevated glucose levels, a decline in the expression of *Gpx1* and *Gpx4* was observed in renal-medulla cells, potentially attributable to distinct antioxidative mechanisms inherent to these cell types. In contrast to renal-medulla cells, renal-cortex cells exhibited a more vigorous and intricate antioxidation process, enabling them to more efficiently counteract the oxidative challenges induced by high-glucose stress. Supranee Ruenkoed et al. observed in their research on Nile tilapia that dietary supplementation with GABA led to heightened expression of antioxidant enzyme genes [26]. Similarly, Mokhtar Fathi et al. demonstrated that incorporating GABA into broiler diets could mitigate oxidative stress triggered by dexamethasone and enhance the activity of antioxidant enzymes such as *CAT*, *SOD*, and *Gpx* [27]. This study, in alignment with prior investigations, validates the antioxidative properties of GABA in animals and elucidates its specific regulatory mechanisms under conditions of high-glucose stress, as well as the divergent responses exhibited by different cell types.

Oxidative stress induces alterations in metabolic products within the body, prompting increased interest in the interplay between oxidative stress and metabolomics. The kidney, a highly active organ rich in mitochondria, is particularly vulnerable to oxidative stress resulting from elevated blood-glucose levels, a factor known to accelerate the progression of chronic kidney disease. Individuals in advanced stages of chronic kidney disease often exhibit indications of oxidative stress [28], which is associated with hypertension and other comorbidities. Consequently, various antioxidant medications are being explored as potential therapeutic interventions for adult patients with chronic kidney disease. A recent investigation conducted antioxidant gene testing on kidney cells from Mongolian sheep, revealing that the addition of GABA to high-sugar environments reduced ROS production, upregulated the expression of antioxidant genes, and potentially facilitated metabolic processes such as glycolysis, the tricarboxylic acid cycle, and the respiratory chain. Employing non-targeted metabolomics, this study scrutinized the metabolic shifts in Mongolian sheep kidney cells under varying sugar concentrations and GABA treatment, aiming to elucidate the impact of GABA on kidney-cell sugar metabolism and antioxidant function. Notably, in renal-cortex cells, upregulation of metabolites such as glucose-6-phosphate, fructose-6-phosphate, glyceraldehyde-3-phosphate, glucose-6-phosphate dehydrogenase, and ribose-5-phosphate was observed in comparison to the high-sugar control group, while L-glutamate levels were downregulated. Furthermore, compared to the normal-sugar experimental group, citric acid and isocitric acid levels were elevated, whereas glutamic acid levels were diminished. Hence, it is hypothesized that GABA could potentially penetrate renal-cortex cells via the transporter protein encoded by the *SLC6A1* gene, facilitating glucose uptake and metabolism. This process leads to the enhancement of intermediary compounds in glycolysis and the tricarboxylic acid cycle (e.g., glucose-6-phosphate, fructose-6-phosphate, glyceraldehyde-3-phosphate, citric acid, and isocitric acid), thus indicating an acceleration of these metabolic pathways. This acceleration could elevate NADH production and promote the formation of ROS through the respiratory chain. As a result, intracellular antioxidant mechanisms are activated, with *SOD1* converting ROS to H_2_O_2_, *CAT* decomposing H_2_O_2_ into its constituent gases, water and oxygen [29], and *Gpx4* further breaking down ROS to water, collectively mitigating oxidative stress within cells. In medullary cells, the high-sugar experimental group exhibited decreased levels of glutamate compared to the high-sugar control group. Additionally, in comparison to the normal-sugar experimental group, elevated levels of glucose-6-phosphate and reduced levels of phosphoenolpyruvate were detected in the high-sugar experimental group. Glucose-6-phosphate is derived from glucose-6-phosphate through a series of reactions, with the NADPH generated in this process playing a critical role in maintaining cellular reduced glutathione (GSH) levels. Therefore, the upregulation of these metabolites may confer antioxidant properties by preserving GSH levels.

This research study validated that treatment with GABA significantly altered metabolites in renal-cortex cells, particularly by elevating various compounds within the glycolysis pathway, suggesting heightened activities of glycolysis and the tricarboxylic acid cycle. The introduction of GABA expedited these metabolic processes, activating antioxidant genes and thereby indicating its potential to mitigate oxidative stress. Additionally, the downregulation of glutamate in both renal-cortex and medulla cells, given that glutamate can be converted to GABA, implies a potential role for glutamate–GABA conversion. Tang Yunfeng et al. discovered that icariin (ICT) treatment notably ameliorated neurological impairments in brain ischemia-reperfusion mice, enhancing antioxidant enzyme activity and upregulating antioxidant-related proteins. Metabolomics analysis revealed that ICT influenced multiple metabolic pathways [30]. Other studies have also shown through metabolomics analysis that berberine (Cop) can diminish inflammatory mediators such as prostaglandin D2 (PGD2) and tumor necrosis factor-alpha (TNF-α) while augmenting SOD expression [31]. Tana et al. further identified through non-targeted metabolomics that Eucommia ulmoides extract substantially reduced body weight and increased levels of antioxidant enzymes, including *SOD*, *CAT*, and *Gpx* [32]. The combined outcomes from these studies and the present research indicate that glucose addition escalates ROS production, whereas GABA supplementation enhances antioxidant enzyme expression and exerts similar effects on substance metabolism.

The results of ROS detection indicated that, in the presence of normal glucose levels, the ROS content increased regardless of the GABA concentration. However, when GABA was introduced under conditions of high glucose, a notable decrease in ROS content was observed at GABA concentrations of 5 mmol/L and 10 mmol/L, suggesting a protective effect of GABA at these levels against oxidative stress induced by high glucose. Conversely, at a GABA concentration of 15 mmol/L, the ROS content exhibited an upward trend, indicating that excessively high GABA concentrations did not further mitigate ROS production. This underscores the importance of carefully determining the optimal GABA concentration to effectively reduce ROS levels without adverse consequences. Interestingly, in this investigation, the addition of varying GABA concentrations to normal glucose levels led to an increase in ROS production alongside enhanced cell viability, contrary to the anticipated cell-damaging effects of ROS. This unexpected outcome may be attributed to methodological constraints or variations in reagent batches. Future research endeavors could employ alternative detection techniques and rigorously control factors such as reagent consistency to corroborate these findings.

In summary, this research examines the potential impact of GABA on the glucose metabolism and antioxidant capabilities of kidney cells from Mongolian sheep. These results deepen our understanding of the metabolic adaptive responses of the kidney in various physiological and pathological states, offering novel perspectives for addressing conditions stemming from oxidative irregularities.

## 4. Materials and Methods

### 4.1. Isolation and Culture of Renal-Cortex and Renal-Medulla Cells from Mongolian Sheep

Mongolian sheep kidney samples were promptly transferred to the cell-culture facility within a timeframe of 4 h. Subsequently, the samples underwent a series of procedures, including washing and division into the renal cortex and medulla using D-phosphate-buffered saline (D-PBS) (BI Company, Herzliya, Israel) supplemented with 1% penicillin–streptomycin (PS) (BI Company, Israel). After the removal of blood, the samples were thoroughly washed and disinfected. The tissue blocks were then fragmented into pieces smaller than 1 mm^3^ and immersed in a complete culture medium containing 10% fetal bovine serum (FBS) (ThermoFisher Scientific, Waltham, MA, USA) and 0.1% penicillin–streptomycin. The cultures were maintained at a temperature of 37 °C in an environment with 5% CO_2_. The culture medium was refreshed every 48 h, and once a confluence of 70–80% was reached, the cells were detached using trypsin (ThermoFisher Scientific, USA), counted after centrifugation, and subsequently seeded into flasks at a concentration ranging from 200,000 to 300,000 cells per flask.

### 4.2. Experimental Grouping According to GABA Concentrations

(1) The normal control group (5.5 + 0): sugar normal culture medium, glucose content is tendency for 5.5/L, excluding GABA; (2) High-glucose control group (25 + 0): medium containing high glucose (25 mmol/L) without GABA; (3) GABA-treatment group (5.5 + 5 tendency, + 10 tendency for 5.5 L and 5.5 L + 15 tendency for L, 25 L, 25 + 5 + 10 tendency, 25 + 15 tendency for L/L).

### 4.3. Assessment of Cell Viability and ROS Levels

After a 24 h period of standard cell culture, we rinsed the cells with 1 mL of D-PBS solution containing 1% PS to eliminate any remaining culture medium. Subsequently, the cells were transferred separately into regular glucose and high-glucose culture mediums and incubated for 48 h at 37 °C with 5% CO_2_. Before replacing the culture mediums, the cells were rinsed once more with D-PBS. Then, we transferred each set of cells into either regular-glucose or high-glucose culture mediums supplemented with varying concentrations of GABA and maintained them under the same conditions for an additional 48 h. A 1000-fold diluted DCFH-DA probe was introduced, shaken gently for 4 h, and we used a laser confocal microscope (ZEISS, Jena, Germany) to assess fluorescence intensity. Following this, 20 μL of CCK-8 solution (biosharp, Hefei, China) was added and incubated for 4 h, and we used an enzyme immunoassay analyzer (Molecular Devices, San Jose, CA, USA) to determine absorbance at 450 nm.

### 4.4. Gene Expression Analysis Using qRT-PCR

Based on the GABA concentrations obtained in the previous step, kidney cells from Mongolian sheep were subjected to treatment for detecting the expression of genes such as antioxidant enzyme genes *CAT*, *SOD1*, *SOD2*, *Gpx1*, *Gpx4*, and genes involved in GABA and glucose transport, including *SLC6A1* and *Glut1*. Please refer to the primer design list in Appendix A.

### 4.5. Sample Preparation for Metabolomics Analysis

Each group was established with three replicates, and the cell count in each replicate ranged from 1 × 10^7^ to 1.5 × 10^7^. The sample was retrieved from a −80 °C freezer and thawed on ice (all subsequent procedures were conducted on ice). Subsequently, 500 μL of internal standard extractant in 80% methanol water (Merck, Darmstadt, Germany) was added, vortexed for 3 min to completely suspend the sample, placed in liquid nitrogen for 5 min, then thawed on dry ice for another 5 min before being transferred back to the ice for an additional 5 min. The mixture underwent vortexing for 2 min, followed by three cycles of freezing in liquid nitrogen, thawing, and vortexing. After centrifugation at 12,000 r/min at 4 °C for 10 min, a volume of 300 μL of the supernatant was transferred into a centrifuge tube and stored in a −20 °C freezer for half an hour before undergoing further centrifugation at the same speed and temperature conditions for an additional duration of three minutes. Finally, a volume of 200 μL of the supernatant was extracted into the corresponding injection bottle vial liner for instrumental analysis.

### 4.6. Chromatography–Mass Spectrometry (CMS) Conditions for Data Acquisition

The chromatography–mass spectrometry conditions are detailed in Appendix A, including the use of a Waters ACQUITY Premier HSS T3 Column (1.8 μm, 2.1 mm × 100 mm), with mobile phase A consisting of 0.1% formic acid (Aladdin, Shanghai, China)/water and mobile phase B containing 0.1% formic acid/acetonitrile (Merck, Germany). The column temperature was maintained at 40 °C, with a flow rate of 0.4 mL/min and a sample quantity of 4 μL. Mass-spectrometry conditions can be found in Appendix A.

### 4.7. Screening of Differential Metabolites

The criteria of VIP > 1 and *p*-value < 0.05 were considered significant differences.

### 4.8. Data Analysis and Statistical Methods

All experimental data were expressed in the form of mean ± standard deviation and processed by SPSS (v20.0) statistical analysis software. The *t*-test method was used to calculate *p* values of the data of each group compared with the control group. *p* < 0.05 indicated statistical significance.

## 5. Conclusions

When Mongolian sheep kidney cells are cultured in a high-sugar environment, they undergo oxidative stress reactions. However, the presence of GABA can mitigate the detrimental effects of high sugar levels by enhancing glycolysis and facilitating the TCA cycle while concurrently upregulating the expression of antioxidant genes, thereby reducing oxidative damage in the cells.

## Figures and Tables

**Figure 1 ijms-25-10033-f001:**
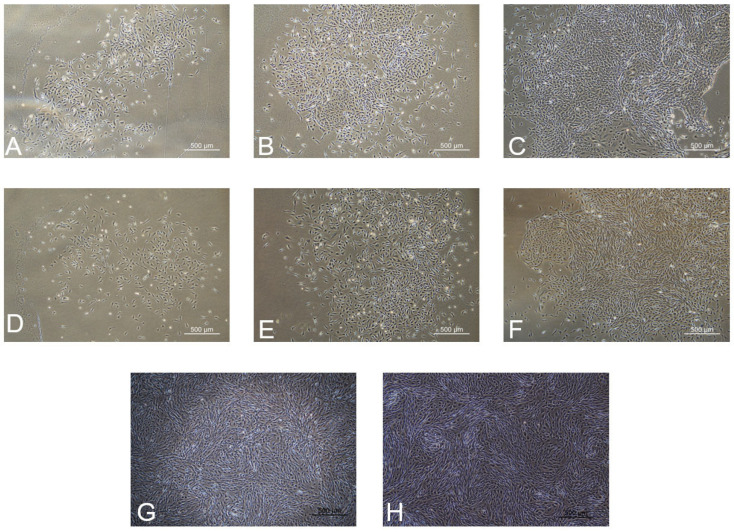
Renal-cortical and medullary cells of Mongolian sheep. (**A**) Renal-cortex primary cell culture 3 d, (**B**) renal-cortex primary cell culture 5 d, (**C**) primary renal-cortex cells cultured for 7 days (20×). (**D**) Renal-medullary primary cell culture 3 d, (**E**) renal-medullary primary cell culture 5 d, (**F**) renal-medullary primary cells cultured for 7 days (20×). (**G**) P3 renal-cortical cells culture for 96 h, (**H**) P3 renal-medulla cells culture for 96 h (20×).

**Figure 2 ijms-25-10033-f002:**
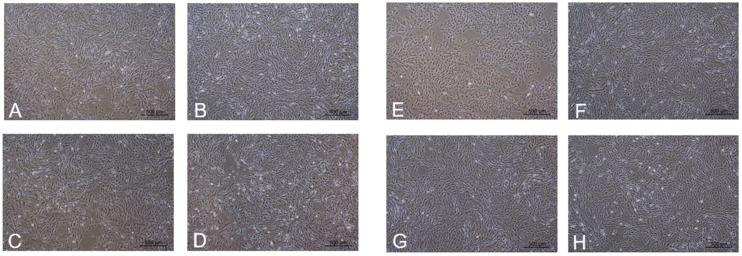
Normal sugar and high-sugar concentrations, adding different concentrations of GABA-cultured renal-cell culture. Normal sugar is added to kidney cells with different concentrations of GABA. (**A**) 0 mmol/L GABA, (**B**) 5 mmol/L GABA, (**C**) 10 mmol/L GABA, (**D**) 15 mmol/L GABA (20×). High sugar adds different concentrations of GABA to kidney cells. (**E**) 0 mmol/L GABA, (**F**) 5 mmol/L GABA, (**G**) 10 mmol/L GABA, (**H**) 15 mmol/L GABA (20×).

**Figure 3 ijms-25-10033-f003:**
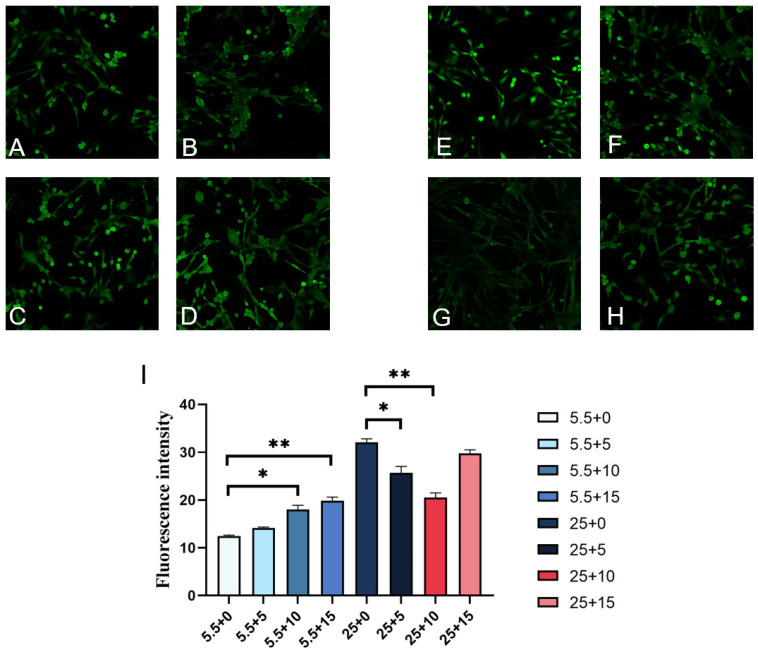
Normal added sugar and high-sugar concentration cultures of different concentrations of GABA-cultured renal-cell ROS fluorescence detection. (**A**–**D**) Fluorescent images of ROS in kidney cells with different concentrations of GABA added to normal sugar. (**E**–**H**) Fluorescence of ROS in renal cells with different concentrations of GABA supplemented with high sugar. GABA concentrations were 0 mmol/L, 5 mmol/L, 10 mmol/L, 15 mmol/L (10×). (**I**) The mean fluorescence intensity (ROS) values of kidney cells with different concentrations of GABA added to high sugar. 5.5. Normal sugar added with different concentrations of GABA, 25. High sugar added with different concentrations of GABA. In the figure, * represents a significant difference (*p* < 0.05), ** represents an extremely significant difference (*p* < 0.01).

**Figure 4 ijms-25-10033-f004:**
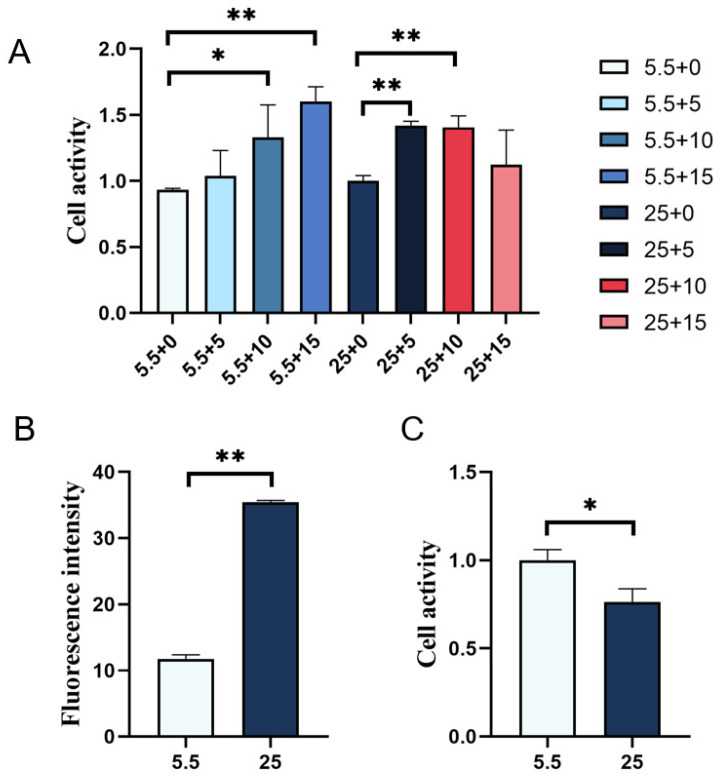
Mean fluorescence intensity (ROS) and cell viability of renal cells cultured in normal glucose and high glucose were detected. (**A**) Normal sugar and high sugar added different concentrations of GABA renal cell viability. (**B**) Mean fluorescence intensity (ROS), (**C**) mean cell viability, * indicates significant (*p* < 0.05), ** indicates extremely significant (*p* < 0.01).

**Figure 5 ijms-25-10033-f005:**
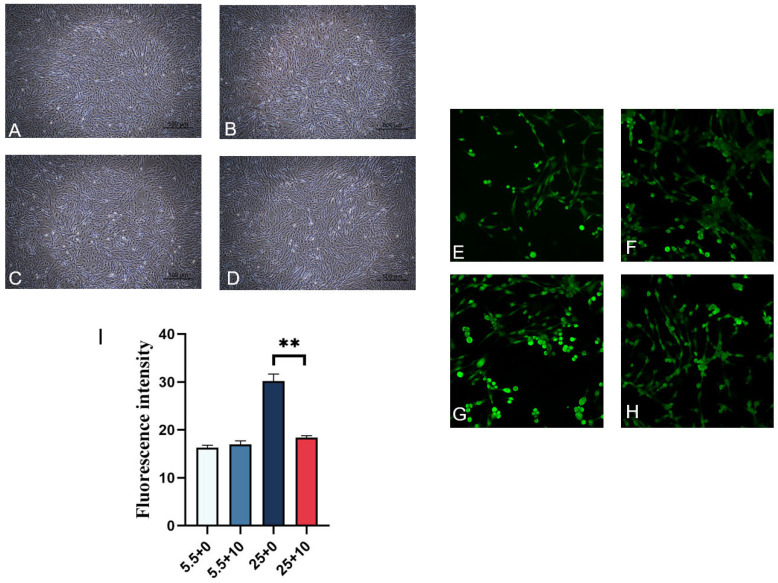
Renal-cortical cells were cultured with normal glucose and high glucose supplemented with GABA. (**A**) Normal sugar does not add GABA, (**B**) normal sugar adds 10 mmol/LGABA, (**C**) GABA is not added for high sugar, (**D**) 10 mmol/LGABA is added for high sugar (20×). (**E**–**H**) Fluorescent images of renal-cortical cells with different sugars added GABA. (**E**) Normal sugar does not add GABA, (**F**) normal sugar adds 10 mmol/LGABA, (**G**) GABA is not added for high sugar, (**H**) 10 mmol/LGABA is added for high sugar (20×). (**I**) Each group’s average fluorescence value, ** represents an extremely significant difference (*p* < 0.01).

**Figure 6 ijms-25-10033-f006:**
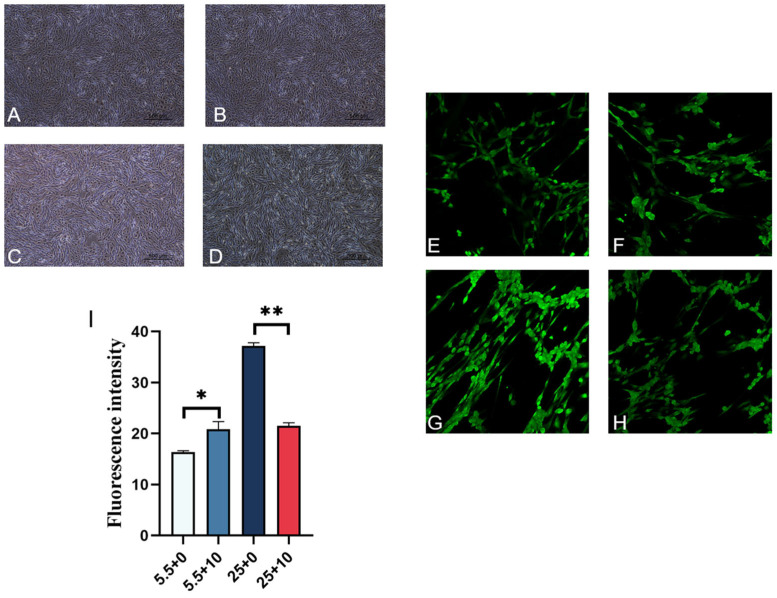
Normal sugar and sugar add GABA in cultivated renal-medulla cells. Renal-medullary cells with different sugars added GABA. (**A**) Normal sugar does not add GABA, (**B**) normal sugar adds 10 mmol/LGABA, (**C**) GABA is not added for high sugar, (**D**) 10 mmol/LGABA is added for high sugar (20×). (**E**–**H**) Fluorescent images of renal-medullary cells with different sugars added GABA. (**E**) Normal sugar does not add GABA, (**F**) normal sugar adds 10 mmol/LGABA, (**G**) GABA is not added for high sugar, (**H**) 10 mmol/LGABA is added for high sugar (20×). (**I**) Each group’s average fluorescence value, In the figure, * represents a significant difference (*p* < 0.05), ** represents an extremely significant difference (*p* < 0.01).

**Figure 7 ijms-25-10033-f007:**
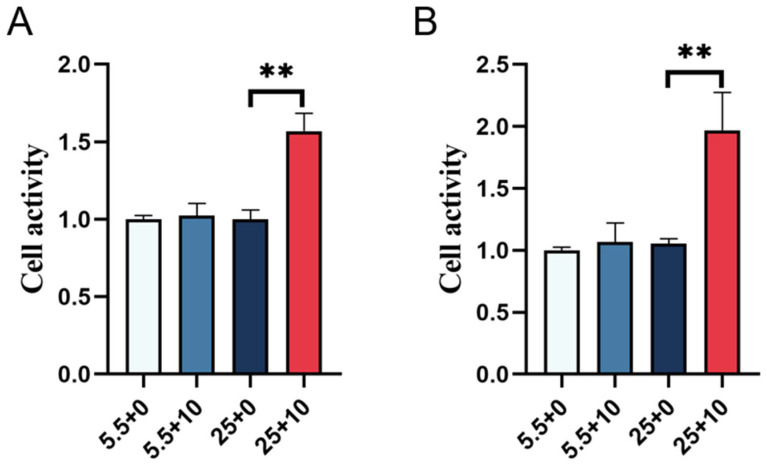
The cell viability values of renal-cortical and medulla cells were measured by adding 10 mmol/LGABA to normal and high glucose. (**A**) Cell viability values of renal cortical cells, (**B**) cell viability values of renal-medulla cells. In the figure, ** represents an extremely significant difference (*p* < 0.01).

**Figure 8 ijms-25-10033-f008:**
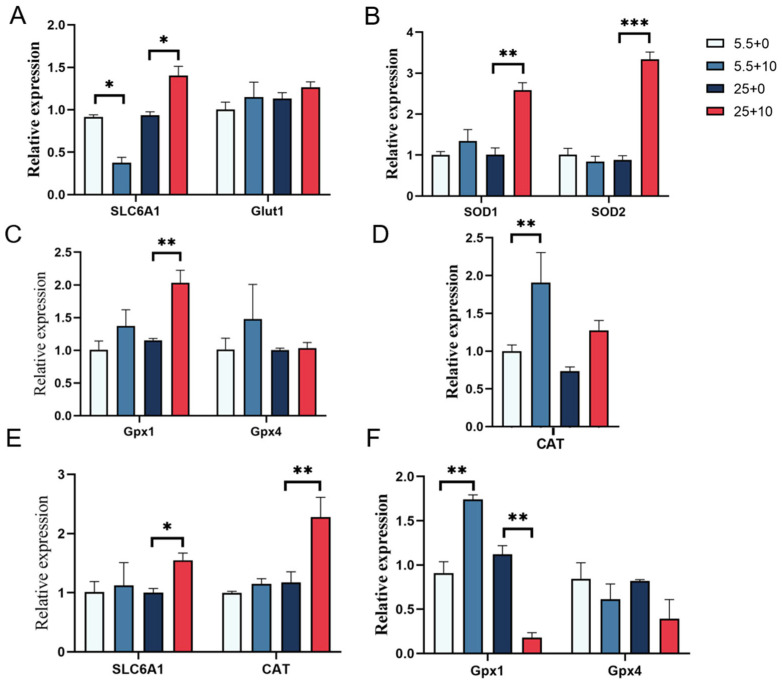
(**A**–**D**) gene expression changes in renal cortex; (**E**,**F**) renal-medullary gene expression changes. In the figure, * represents a significant difference (*p* < 0.05), ** represents an extremely significant difference (*p* < 0.01), *** represents an extremely significant difference (*p* < 0.001).

**Figure 9 ijms-25-10033-f009:**
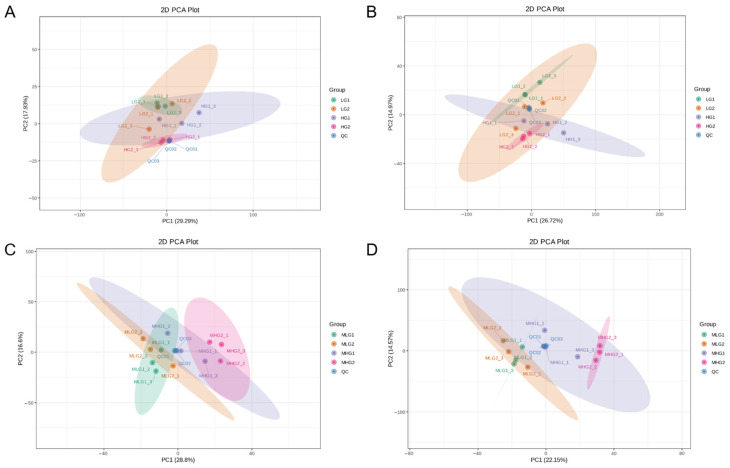
Principal component analysis diagram of the total sample of Mongolian sheep renal-cortex and renal-medulla cells. (**A**,**B**) of renal-cortical cells; (**C**,**D**) of renal-medulla cells.

**Figure 10 ijms-25-10033-f010:**
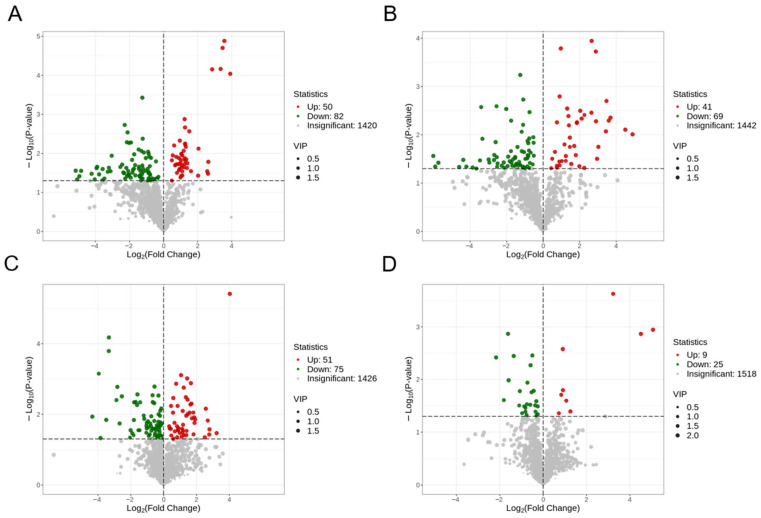
Volcano map of differential metabolites in renal-cortical cells of Mongolian sheep. (**A**) HG2_vs_HG1, (**B**) LG1_vs_HG1, (**C**) LG2_vs_HG2, (**D**) LG2_vs_LG1. In the volcano plot, each dot represents a metabolite. The green dot indicates a downregulated differential metabolite, the red dot indicates an upregulated differential metabolite, and the gray dots represent detected but not significantly different metabolites. The x-axis shows the absolute value of the numerical difference in relative content (log_2_FC) for multiple metabolites between two groups of samples, with larger values indicating greater differences in relative content between the two groups. The y-axis represents the level of significance (−log10*p*-value), and the size of the dots reflects their VIP value.

**Figure 11 ijms-25-10033-f011:**
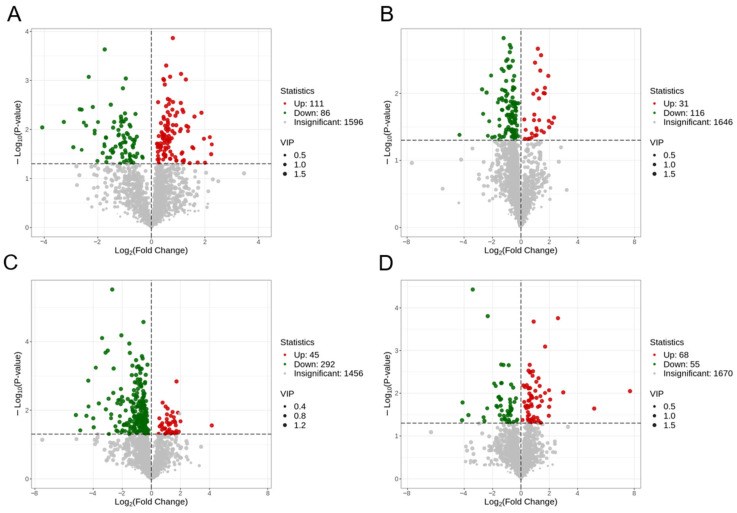
Volcanic map of differential metabolites in renal-medullary cells of Mongolian sheep. (**A**) MHG2_vs_MHG1, (**B**) MLG1_vs_MHG1, (**C**) MLG2_vs_MHG2, (**D**) MLG2_vs_MLG1.

**Figure 12 ijms-25-10033-f012:**
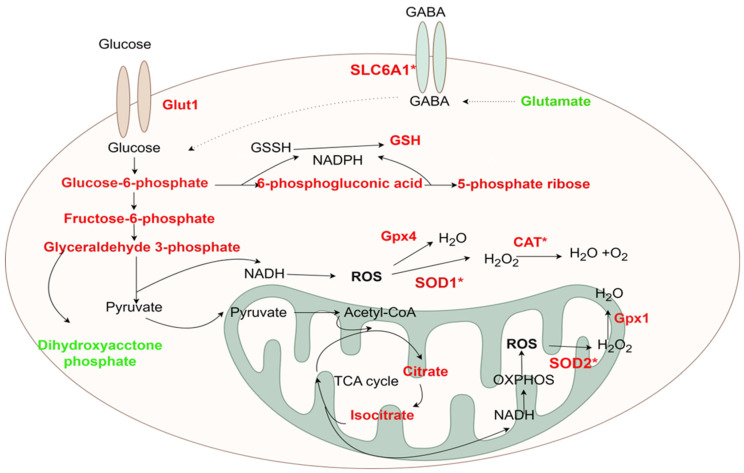
Antioxidant regulation network of GABA in the kidney cells. Bold red: Up-regulated metabolites; Bold red band *: Significantly increased genes; Bold green: Down-Regulated metabolites. Solid lines represent direct effects and dashed lines represent po-tential or possible effects. By Figdraw.

## Data Availability

Metabolomics data have been deposited to the EMBL-EBI MetaboLights database (https://doi.org/10.1093/nar/gkad1045, PMID: 37971328) with the identifier MTBLS10655.

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
