# Peer review of "Effects of GABA on Oxidative Stress and Metabolism in High-Glucose Cultured Mongolian Sheep Kidney Cells"

_ijms, 2024, doi:10.3390/ijms251810033_

Round 1
Reviewer 1 Report
Comments and Suggestions for Authors
Reviewer’s comments on the manuscript by Su et al. entitled: Effects of GABA on Oxidative Stress and Metabolism in High-Glucose Cultured Mongolian Sheep Kidney Cells.
Manuscript ID: ijms-3146701
July, 2024.
General Comments
The manuscript needs to be revised before publication in the Journal. Specific Comments that are listed for the different section below:
Specific Comments
L428-447: I don't know what is the meaning of this discussion? How about the references also, please rewritten.
L450: please added the detail of sheep numbers, sex, and body weight, because these factors can effect on the results.
L462-467: How about the sample size? Please added how many sample size in this study. Moreover, please explained why added GABA levels? How about the references support?
L509-513: how to calculate the relative gene expression data, the authors need to added the detail steps in this Section. Moreover, the author analysis the data using the t-test, the authors need to explain why using the t-test? Why the authors not use the ANOVA? Thirdly, the Metabolomics data also using the t-test? need to clear in this Section!
L514-519: please rewritten the Conclusion section! Please truthfully state your findings, such as, “When Mongolian sheep kidney cells are cultured in a high-sugar environment, they undergo oxidative stress reactions”, Are you sure this description is appropriate?
L538: Please rewrite this section content carefully!!!!!!
Reviewer 2 Report
Comments and Suggestions for Authors
The introduction should be more concise and directly relevant to the study's findings. This will help set the stage for the research and maintain the reader's focus.
The discussion section is currently difficult to follow due to its broad scope. To improve clarity and impact, it should be more concise and focused. Consider simplifying some sections or providing clearer explanations to make the discussion more accessible. Structuring the discussion more clearly—starting with a summary of the main findings, followed by comparisons with existing literature, implications and future directions—would be beneficial. Although some references are mentioned, the discussion could be enhanced by a more thorough comparison with existing studies, which would help highlight their significance.
Reviewer 3 Report
Comments and Suggestions for Authors
The manuscript ijms-3146701 entitled "Effects of GABA on Oxidative Stress and Metabolism in High-Glucose Cultured Mongolian Sheep Kidney Cells" investigates how GABA mitigates oxidative stress and influences metabolism in high-glucose cultured Mongolian sheep kidney cells, revealing potential protective mechanisms that could enhance stress resilience. The authors should consider the following points:
1. Statistical Analysis and Metabolomics:
1-1. Please include information on whether the data were checked for normal distribution to justify the use of parametric statistical methods.
1-2. It is unclear what type of ANOVA was used. For this type of data, two-way ANOVA should be applied (with sugar factor: normal sugar vs. high sugar, GABA factor: vehicle vs. GABA concentrations, and interactions between these factors) in Figures 3I, 4A, 5I, 6I, 7A-B, and 8A-F.
1-3. Please include a detailed statistical report (F, df, exact p-values) for main effects and interactions. Additionally, the results of post-hoc analyses should be discussed to clarify the differences between the experimental groups.
2. Data and Figures:
2-1. Metabolomics data are included, but the authors should provide a more detailed discussion and supplementary data showing the key metabolites altered by GABA treatment and their specific roles in oxidative stress and metabolic regulation.
2-2. The study reports that 15 mmol/L of GABA increases cell death under normal glucose conditions. Could the authors discuss potential reasons for this cytotoxic effect at higher GABA concentrations?
2-3. The results show a different response of renal cortex and medulla cells to GABA treatment. What might account for these differences, and how could they influence the overall interpretation of GABA’s protective role?
2-4. Please add scale bars to the microscopic images.
Round 2
Reviewer 1 Report
Comments and Suggestions for Authors
The authors have been changed the manuscript according to my suggestion. and the revised manuscript can be accepted for publication.
Reviewer 3 Report
Comments and Suggestions for Authors
Everything is okay.